## Foot and ankle Osteoarthritis and Cognitive impairment in retired UK Soccer players (FOCUS): protocol for a cross-sectional comparative study with general population controls

Shima Espahbodi [ID],[1,2] Gwen Fernandes [ID],[1,3] Eef Hogervorst,[4] Ahmed Thanoon,[1,2] Mark Batt,[2,5] Colin W Fuller,[6] Gordon Fuller [ID],[7] Eamonn Ferguson,[8] Tobias Bast,[8,9] Michael Doherty,[1,2] Weiya Zhang[1,2]

**Correspondence to**
Dr Shima Espahbodi;
shima.espahbodi1@nottingham.ac.uk

## ABSTRACT

**Introduction** Professional footballers commonly experience sports-related injury and repetitive microtrauma to the foot and ankle, placing them at risk of subsequent chronic pain and osteoarthritis (OA) of the foot and ankle. Similarly, repeated heading of the ball, head/neck injuries and concussion have been implicated in later development of neurodegenerative diseases such as dementia. A recent retrospective study found that death from neurodegenerative diseases was higher among former professional soccer players compared with age matched controls. However, well-designed lifetime studies are still needed to provide evidence regarding the prevalence of these conditions and their associated risk factors in retired professional football players compared with the general male population.

**Objectives** To determine whether former professional male footballers have a higher prevalence than the general male population of: (1) foot/ankle pain and radiographic OA; and (2) cognitive and motor impairments associated with dementia and Parkinson's disease. Secondary objectives are to identify specific football-related risk factors such as head impact/concussion for neurodegenerative conditions and foot/ankle injuries for chronic foot/ankle pain and OA.

**Methods and analysis** This is a cross-sectional, comparative study involving a questionnaire survey with subsamples of responders being assessed for cognitive function by telephone assessment, and foot/ankle OA by radiographic examination. A sample of 900 adult, male, ex professional footballers will be recruited and compared with a control group of 1100 age-matched general population men between 40 and 100 years old. Prevalence will be estimated per group. Poisson regression will be performed to determine prevalence ratio between the populations and logistic regression will be used to examine risk factors associated with each condition in footballers.

**Ethics and dissemination** This study was approved by the East Midlands-Leicester Central Research Ethics Committee on 23 January 2020 (REC ref: 19/EM/0354). The study results will be disseminated at national and international meetings and submitted for peer-review publication.

### Strengths and limitations of this study

► Largest study of head injuries/heading, concussion, ball type and cognitive impairment in retired professional footballers versus controls in the UK.
► Largest study of foot/ankle injuries and osteoarthritis/pain in retired professional footballers versus controls in the UK, with standardised foot/ankle radiographs obtained in a sample of each.
► Case-control study design with control men recruited from a community-based population sample representative of men in the UK general population.
► Self-reporting of injuries and concussive symptomatology during careers/lifetime may be subject to recall bias, and non-involvement by individuals with severe cognitive impairment may cause left censorship.

## INTRODUCTION

Professional footballers are at high risk of sports-related injuries, including those to the head, thigh, knee and foot/ankle.[1] Over the past 25 years, the prevalence of injuries to the thigh (23%–26%), foot/ankle (20%–23%) and knee (14%–18%) has consistently been reported to be the three most common injury sites in professional football.[2–6] A study of one English Premier League club found that over a 4-season period, 20% of all injuries involved the foot/ankle with a mean return to play time of 54 days.[7] Injuries to the ankle lateral ligament complex, particularly the anterior talofibular ligament, were most frequent, accounting for 31% of all foot/ankle injuries. More minor repetitive impact-loading to the ankle/foot may also lead to joint and

periarticular injury. However, despite the high incidence of acute injuries, there is a paucity of robust evidence regarding subsequent osteoarthritis (OA) development and chronic pain in the foot/ankle joints of professional footballers. Some studies report a prevalence of ankle OA ranging from 12% to 17%.[1 8] However, these studies had small sample sizes without adequate control groups and were based on self-reported diagnosed ankle OA. Importantly, there are no published data on the prevalence of foot pain and foot OA in retired professional footballers. Consequently, foot and ankle OA is not formally recognised as an occupational disease for professional footballers, and there is a lack of evidence regarding the football-related risk factors associated with this condition.

Retired professional footballers with OA have a significantly lower health-related quality of life (QoL) compared with players without OA.[9] Ninety per cent of former professional footballers suffering with OA have moderate to severe joint pain, and 37% of these retired players reported moderate to severe depression and anxiety due to their medical condition.[1 10] In ex-UK footballers with OA, a higher frequency of disability and work-related disability has been reported compared to those without OA.[11] More recently, we reported that depressive and anxiety symptoms in ex-footballers are comparable to that in the general population. However, ex-footballers reported significantly more sleep problems, negative mood profiles, more widespread bodily pain and higher analgesic usage compared with controls.[12] While OA might not be a life-threatening disease, its long-term effects on both the physical and mental health of a retired footballing population need further attention.

Professional footballers may also be at increased risk of developing neurodegenerative diseases such as dementias and Parkinson's disease, potentially arising from major neck/head injury, overt mild traumatic brain injury (mTBI) causing concussion, or repetitive micro-trauma from heading the ball.[13–16] TBI has been identified as a risk factor for the development of dementia in the general population, specifically Alzheimer's, in a longitudinal cohort study (n=2.8 million) spanning 36 years.[17] In that study, moderate to severe TBI increased dementia risk across all ages, whereas mTBI (concussion) increased risk in those aged over 65 years.[17] Repetitive heading of the ball during matches and training sessions, head-to-head or head to elbow/knee/foot collisions between players, or accidental heading might additionally result in concussion, neuropsychiatric and cognitive deficits.[18] The recent FIELD study retrospectively compared mortality from neurodegenerative disease in 7676 former professional soccer players with that among 23 000 matched general population controls in Scotland. Former footballers had a 3.5 times higher death rate from neurodegenerative diseases compared with matched controls. Risk of death varied according to disease subtype and was five times higher in those with Alzheimer's and two times higher in those with Parkinson's, respectively.[19] While of considerable importance, these findings need

to be confirmed in the lifetime of professional footballers and matched controls. Still, little is known about the increased prevalence of neurodegenerative diseases and related risk factors among retired professional players. Recent literature continues to be conflicting and no study has provided conclusive evidence for the relationship between heading/head impacts and neurodegenerative diseases. Two recent systematic reviews support these conclusions and emphasise the need for high quality, large-scale prospective studies.[20 21]

Previously, we demonstrated a 2–3 times higher prevalence of knee OA in 1207 male ex-footballers compared with 4085 men in the general population[22] and identified knee injury and training load as the main football-specific factors for this.[23] We will use the same population samples for the Foot/ankle Osteoarthritis and Cognitive impairment in UK Soccer players (FOCUS) study. The primary objectives of this study are to determine whether or not, compared with age-matched men in the general population, ex-professional UK footballers have a higher prevalence of:

1. foot/ankle pain and radiographic OA; and
2. cognitive impairment, and neurodegenerative diseases particularly dementia and Parkinson's disease.

Secondary objectives are to identify factors associated with the risks of these conditions in footballers.

We have intentionally included investigations into foot/ankle OA together with neurodegenerative disease in this single cross-sectional study as it will allow both important research questions to be addressed concurrently. Furthermore, we anticipate that embedding questions and assessments on mental/cognitive function into a general health survey may increase the response rate and minimise the selection bias compared with asking questions in a survey focused solely on a single disease.

## METHODS AND ANALYSIS
### Study design
This will be a cross-sectional study comparing UK male ex-professional footballers and age-matched men in the general population. It will involve three discrete stages: (1) a postal questionnaire survey to all study participants; (2) a telephone assessment of cognitive function in a subsample of each group; and (3) a radiographic assessment of both feet and ankles in a subsample of each group. The planned start and end dates for the study are August 2020–December 2022. Figure 1 shows the study stages and procedures.

### Participants and recruitment
The FOCUS footballer questionnaire, together with a participant information sheet, consent form, stamped return envelope, and letter of endorsement from the Professional Footballers' Association (PFA) and Football Association (FA), will be posted across the UK to 900 ex-professional male footballers who previously participated in the Nottingham knee OA footballer study[22 24] and

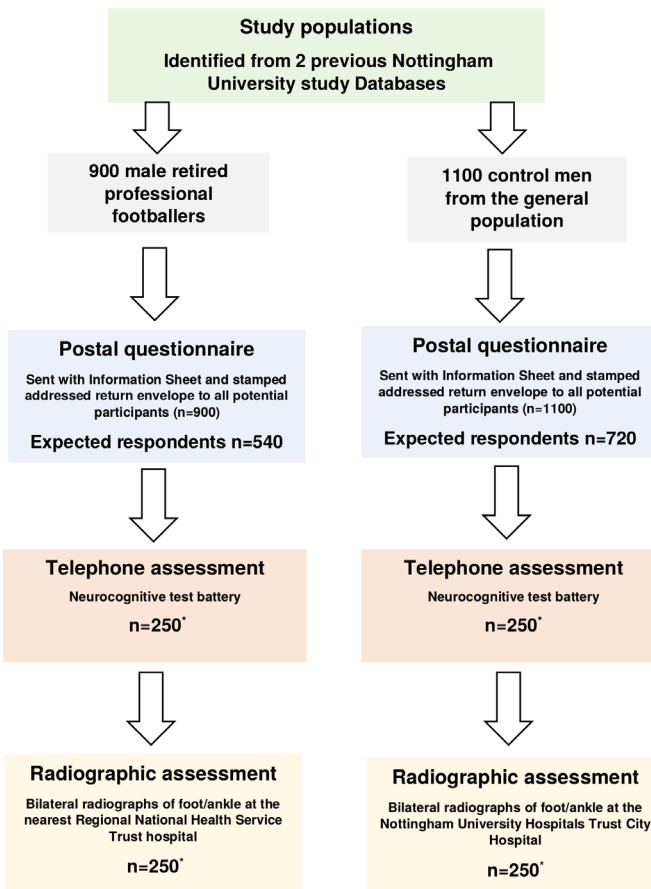

**Study populations**

Identified from 2 previous Nottingham University study Databases

900 male retired professional footballers

1100 control men from the general population

**Postal questionnaire**

Sent with Information Sheet and stamped addressed return envelope to all potential participants (n=900)

**Expected respondents n=540**

**Postal questionnaire**

Sent with Information Sheet and stamped addressed return envelope to all potential participants (n=1100)

**Expected respondents n=720**

**Telephone assessment**

Neurocognitive test battery

n=250*

**Telephone assessment**

Neurocognitive test battery

n=250*

**Radiographic assessment**

Bilateral radiographs of foot/ankle at the nearest Regional National Health Service Trust hospital

n=250*

**Radiographic assessment**

Bilateral radiographs of foot/ankle at the Nottingham University Hospitals Trust City Hospital

n=250*

**Figure 1** Flow chart of Foot/ankle Osteoarthritis and Cognitive impairment in UK Soccer players study stages and procedures. *Minimum sample size according to sample size calculation.

who indicated willingness to be contacted again for future studies. All are retired male footballers aged 40–100 years old, registered with the PFA or club player Association (League Club level). A slightly adapted questionnaire (eg, no questions are included related to professional football), participant information sheet, consent form and stamped return envelope will be sent to 1100 age-matched men aged 40–100 in the East Midlands general population, who are on the Knee Pain in the Community database[25] and have indicated willingness to be contacted about new studies. This is the same source for general population controls as in our previous study on knee OA in footballers.[22] The participant information sheets state the entirely voluntary nature of the study with no inducements for participation. After 4 weeks, a reminder letter with questionnaire pack will be posted to those participants from whom there has been no response to the initial questionnaire pack.

Questionnaire responders who indicate willingness to be contacted further for telephone assessment and/or radiographs will be selected for stages 2 and 3, respectively. At least 250 telephone assessments will be conducted first, and the radiographs will be obtained later when COVID-19 restrictions permit us to use National Health Service (NHS) Trust Radiology facilities. Exclusion

criteria for the assessments include an inability to provide written informed consent (both assessments) and severe injury, amputation or neuropathic (Charcot) arthropathy of both feet (radiographic assessment only), and at the participant's request.

At least 250 footballers will be invited to attend the radiology department of their nearest NHS collaborating hospital where foot/ankle radiographs will be undertaken. The selection will be based on first come first served, as well as the convenience to travel to the nearest hospitals. Currently, we have five collaborating NHS hospitals including Nottingham, Salford, Leeds, Southampton and Imperial College London. Similarly, at least 250 control men from a community-based population sample located within Nottingham and adjacent areas (Derbyshire, Lincolnshire, Leicestershire) representative of men in the general UK population will have radiographs at Nottingham City Hospital campus. Return of completed questionnaires will be taken as consent to the questionnaire survey. For those participants agreeing to a telephone assessment consent will be taken verbally over the phone, after the participants have read and agreed to the consent form and patient information sheet posted to them. All participants attending for radiographs will be met by our research team who will obtain written informed consent prior to the taking of radiographs.

## Questionnaire

Our FOCUS footballer questionnaire was designed to capture detailed information about foot/ankle pain, General Practitioner (GP)-diagnosed OA, significant injuries, surgery and injections to the feet/ankles. Latter sections of the questionnaire capture detailed information about GP-diagnosed dementia, Alzheimer's and Parkinson's disease, the frequency of heading the ball during professional play, the type of ball, concussion history/symptoms, head injuries and memory/cognition problems. For the general population male controls, the questionnaire includes all the same questions except for the absence of specific football-related questions. The layout/format of the questionnaire is consistent between footballers and control men.[25] Table 1 summarises the FOCUS questionnaire domains.

The questionnaire requests information about the individual, their medical history, smoking and alcohol usage, and all current medications whether prescribed or over the counter. A body pain mannequin is used to record pain experienced for most days of the past month anywhere in the body, and the Widespread Pain Index and System Severity Scale[26] are included to identify symptoms and likely diagnosis of fibromyalgia. Line drawings illustrating different regions of the foot (ie, toes, midfoot and heel) allow participants to specify the site of any significant pain for most days in the past month and/or over a 3-month period. The Pain Detect Questionnaire, modified to focus on ankle/foot pain only, will be used to screen for neuropathic type pain,[27] and the participants'

**Table 1** FOCUS questionnaire domains and measurements

| Section domains | Questions and instruments |
|---|---|
| 1. Demographic and current health | Date of birth, height, weight, list of comorbidities including diabetes, stroke, dementia, Alzheimer's, Parkinson's, depression, anxiety, HBP. Smoking and alcohol consumption. All current medication both prescribed and alternative/over the counter. |
| 2. Bodily pain | Body pain mannequin for previous month's pain (current), fibromyalgia mannequin (WPI) and SSSs past week's pain. |
| 3. Foot and ankle pain | Pain present in the feet and/or ankles in the past month (current) and/or past 3 months (chronic), age at pain onset, diagnosis of foot OA, region specific significant foot pain (ankle, mid-foot, big toe, other toes) and age range at onset of pain. Pain Detect Questionnaire and Pain Catastrophizing Scale. |
| 4. Injuries and operations to feet/ankles | Significant football-related injury to ankles/feet/toes. Region specific (ankle, mid-foot, big toe, other toes) total number of significant football injuries throughout career, type of injury (eg, ligament rupture) and age. Significant NON-football related injury ever (same questions as above). |
| | Injections into ankles, type of injection (cortisone, anaesthetic, don't know, or other), greatest number of injections into each ankle in a season, and throughout career. Injections elsewhere in foot, where and how many. Surgical interventions to ankle or toes, type of procedure (eg, fracture fixation) and age. If flat-footed as informed by health professional. |
| 5. Hallux valgus | Current and constitutional toe alignment using line drawings (straight or bunion/degree of angulation). |
| 6. How you feel | Hospital Anxiety and Depression Scale (HADS). |
| 7. Quality of life | Short Form-36 (SF-36) Health Survey. |
| 8. Heading during professional football | Heading frequency during a professional match and training session. General heading play (minor, moderate, major) during a match. Specific 'Only Heading' training sessions ever, frequency pre-season and during season. Type of football played with during career. |
| 9. Concussion and head injuries | Concussion definition. Concussion diagnosis from professional football play, number of episodes, symptoms. Concussion diagnosis from other/accident/fall, number, symptoms. Serious head injury, number, diagnosis. Frequency of Head impact NOT from heading not concussive. Concussion symptoms ever from professional play (unreported), frequency, symptoms. Concussion symptoms ever (unreported) outside of professional football for example, fall, accident, number, symptoms. |
| 10. Memory and thinking | Current Memory problems, worsening, causing difficulty with daily function, work or socially. The Test Your memory (TYM) Questionnaire. |

Pain Detect Questionnaire, Pain Catastrophizing Scale, Hospital Anxiety and Depression Scale, SF-36 Health Survey and Test Your Memory Questionnaire are referenced in the main text.
HBP, high blood pressure; OA, osteoarthritis; SSS, Symptom Severity Scale; WPI, Widespread Pain Index.

thoughts and feelings during their foot pain experience will be measured with the Pain Catastrophizing Scale.[28]

The footballers are asked to report any football-related significant injury ever sustained to their feet, ankles or toes as one which *caused pain for most days for at least a 3-month period* and the type of injury. A similar free text and tick-box table is included which asks for any significant non-football related injuries to the feet/ankles/toes. Information regarding injections into the ankles over the course of the players career is requested via a tick box table (type of injectable, location, side) and total number ever received in any season. A similar table with a free text and tick-box asks the footballers to report any surgery to their ankle and/or toes. Control participants are asked the same set of questions about significant injury, injections and surgery to the ankles and/or toes but without mention of football. Current toe alignment and constitutional toe alignment (in their 20s) will be assessed in all participants using validated line drawings.[29]

Anxiety and depression symptoms will be assessed using the well validated Hospital and Anxiety Depression Scale.[30 31] QoL will be measured with the well-validated RAND 36-item Health Survey 1.0 (SF-36) adapted for the physical and mental health components, respectively.[32 33]

Information related to heading the ball during footballers' professional careers will be captured with detailed questions developed with our retired footballer group (see Patient and public involvement). This will be followed with questions related to concussion diagnoses,

symptomatology and head injuries/impacts both during professional play and outside of professional football.[34] Control participants will be asked the same questions relating to head injury and concussions but without mention of football.

Final sections will capture information on memory/cognition and include the Test Your Memory (TYM) assessment,[35] which can be self-administered for use in postal surveys (personal communication JM Brown 2019). TYM is a short cognitive test for detecting Alzheimer's disease (AD) and other cognitive problems and is reported to be more sensitive for detecting mild AD than the Mini Mental State Examination (MMSE).[35]

**Telephone cognitive assessments**

Telephone assessments will follow a standardised script starting with the researchers introducing themselves and the study, checking that the participant is in a room without distractions, whether they use a hearing aid, confirming consent, their name and date of birth. An initial call will be made to book in the telephone assessment at a time convenient to each participant. The validated tools described below will be administered as per published procedures.

Originally developed as a dementia screen, the Telephone Interview for Cognitive Status-modified (TICS-m) is a test for cognitive function which can be administered over the telephone.[36 37] It includes four domains on (1) orientation; (2) registration (free recall), recent

memory and delayed recall (memory); (3) attention/calculation; (4) comprehension, semantic memory and repetition (language). The 13-items (maximum score 39) include: (1) day, date, season, age, telephone number; (2) a 10-word list learning exercise then free recall of that word list; (3) subtractions and counting backwards; (4) responsive naming, current reigning monarch and prime minister; (5) word opposites; (6) repetition; and (7) delayed recall of 10-word list. TICS-m is well validated for detecting a range of mild to moderate cognitive disorders with comparable sensitivity and specificity as a screening tool for dementia and Alzheimer's.[38–40]

The Verbal Fluency Test (VFT) is a short task of verbal functioning and typically consists of a task called category (or semantic) fluency where participants are given 1 min to name as many unique words as possible within a semantic category (animals in this study). The category 'animals' is often used; however, alternatives are 'fruits and vegetables', 'cities and towns' and 'items of clothing'. The participant's score is the number of unique correct words produced in 60 s.[41] Word generation deficits in those with dementia, including Alzheimer's, have been well documented.[42]

The Hopkins Verbal Learning Test (HVLT)[43] is a quick and easy to administer tool which will be used to screen for dementia. It is a word-learning test measuring episodic verbal memory, consists of three trials of free-recall of a 12-item (word) list followed by yes/no recognition from the participant. After the third learning trial, the participant is read 24 words and asked to say 'yes' for each word that appeared on the recall list (12 targets) and 'no' for each word that did not (12 distractors). The total immediate recall (reflecting learning ability, is obtained by repeating the same word list three times and adding up all correctly recalled words over the three trials) is calculated. Good validity and reliability have been demonstrated for the HVLT, and it is cross culturally applicable and well tolerated by elderly people.[44]

Along with cognitive measures, we have included the assessment of functional activities as impairment in these is a critical feature in the diagnosis of dementia.[45] Lawton's Instrumental Activities of Daily Living (IADL) Scale is one of the most commonly used tools consisting of 8 items including two community items (mode of transport and shopping), 3 personal management (ability to use the telephone, handle money matters and prepare one's own medication) and 3 household items (doing housekeeping, laundry, and preparing food). Responses to each of the eight items in the scale are coded as 0 (unable), 1 (partially able) or 2 (able) and are the responses summed. Summary scores range from 0 (low function, dependent) to 8 (high function, independent).[46]

## Radiographic assessment

Informed written consent will be obtained prior to radiographs which will be performed by each respective NHS Trust's radiology staff. Radiographs will be taken bilaterally and weight-bearing for the foot and ankle separately.

Foot dorsoplantar, medial oblique, and lateral views, and ankle anteroposterior views will be taken, as detailed in the Standard Operating Procedure. Examination and scoring of region-specific OA will be done of the hindfoot (ankle, subtalar), midfoot and first metatarsophalangeal joint using the La Trobe Foot Atlas and scorings.[47 48] Those with a Kellgren and Lawrence (KL) score of less than 2 for any foot/ankle compartment and equivalent categories (ie, completely normal, possible osteophyte or doubtful narrowing) will be defined as non-OA. Those who satisfy the definition of a KL score of greater than 2 (≥2) or equivalent for any compartment of either foot or ankle will be defined as OA.[49] Also, those with foot/ankle OA and concurrent foot/ankle pain will be defined as symptomatic foot/ankle OA.

## Patient and public involvement

Questionnaire sections specific to professional football play were developed in partnership with ex-professional footballer groups, while those specific to sports-related concussion and memory/cognitive decline were developed in collaboration with concussion and dementia experts (JM Brown).

Our retired player group contributed to the content, design and layout of the FOCUS Questionnaire. Through in-person meetings (SE), two retired professional players from Notts County Football Club directly advised (having had feedback from other retired players) specifically on the questions pertaining to:

► Heading frequency and bandings.
► Training sessions, frequency and types.
► Football types.
► Types of injuries and question structure.
► Head impacts and question content.
► Questions on memory/cognitive function and the TYM test.

The final questionnaire design was reviewed and trialled by 10 retired players at the PFA and 5 from Notts County Football Club. The time range for the ex-players to complete the questionnaire was 30–60 min and the feedback received was positive.

## Outcome measures and statistical analysis

Continuous data will be presented as mean and SD and categorical variables as frequencies and percentages. The statistical tests used to determine whether there is a significant difference between responses received from footballers and controls will be: t-test for normal distributions, Mann-Whitney U test for non-normal distributions and $\chi^2$ test for proportions with exact p values reported. Statistical significance will be set at p<0.05 with no allowances made for the number of statistical tests undertaken.

The prevalence of foot/ankle OA or neurodegenerative disease (and cognitive impairment) (self-reported or clinically assessed) with 95% CI will be estimated per population. A Poisson regression model will be used to calculate prevalence risk ratio, adjusted for confounding factors such as age, body mass index, education and

socioecoomic status (using postcode as a proxy measure to determine the Index of Multiple Deprivation). Linear (for continuous outcome) and logistic (for dichotomous outcome) regression analyses will be used to determine risk factors related to the outcome of interests within footballers. The results of this study will follow the Strengthening the Reporting of Observational Studies in Epidemiology guidelines for the reporting of observational studies.[50]

## Sample size calculation

We used a value of 14.3% prevalence for foot/ankle OA in the general population[51] and an OR of 2 (to be clinically meaningful) between retired footballers and general population. The sample size was calculated using the z test and a multiple logistic regression model with a correlation factor $R^2$ of 0.3 among multiple risk factors/covariates using G*Power V.3.1.9.2.[52] Assuming footballers would have a greater risk of foot/ankle OA than controls (one tail), with power at 90% and significance at 0.05, 210 participants per group is required for foot/ankle OA outcome. We also calculated the sample size for neurodegenerative disease using the same method. According to 6.4% prevalence for neurodegenerative diseases in the general population,[53] and 3.5 times more risk in the footballers reported recently in the FIELD study,[19] the sample size required is 113 per group for this outcome with power at 90% and significance at 0.05. We will therefore use the larger sample size 210 per group for this study. We will recruit at least 250 retired professional footballers and 250 general population control men with a 16% leeway to account for potential dropouts and attrition.

For the postal questionnaire, we will contact all available retired professional footballers (n=900) and an age-matched sample of general population men (n=1100). Considering a 60% response rate based on our previous follow-up study,[54] we would expect 540 footballers and 720 control men to respond to the questionnaire. This will provide sufficient samples from both groups to recruit the subsample of at least 250 participants per group required for the clinical assessments to achieve 90% power.

## DISCUSSION

To the best of our knowledge, this is the largest comparative study aiming to examine the prevalence and associated risk factors of the two common conditions, foot/ankle OA and neurodegenerative disease in retired professional footballers versus age and sex matched general population controls. The study is expected to confirm the recent findings from the FIELD study concerning the increased risk of neurodegenerative disease in professional football players, but will also provide new evidence on whether concussion, head injury, ball type, number of games, training, position of play are related to the increased risk of having this condition in later life. Unlike the FIELD study, this study will recruit participants who are still alive to measure

both severe disease phenotypes such as physician diagnosed dementia or Parkinson's as well as milder phenotypes such as mild cognitive impairment (MCI). Foot/ankle pain and OA will also be measured and compared between the two populations, which will further characterise the distribution of OA and provide further evidence on the burden of foot/ankle OA in professional footballers over the general population controls following our previous study on knee OA.[22] An advantage of this 2 in 1 study is that it may help to mitigate the selection bias, which often occurs in a single disease study where people with the study condition may be more likely (or less likely) to participate. Comorbidities/multimorbidity such as cardiovascular diseases, diabetes, depression, chronic widespread pain, fibromyalgia and QoL will also be collected to compare the general health status between footballers and matched controls.

However, the study has several limitations. First, it is a cross-sectional study which can only confirm association but not causality. We have therefore planned to follow-up the two cohorts for incidence of the diseases in the future. This has been considered for knee OA and multimorbidity following our previous study in knee OA, and we will be able to follow-up the incidence of neurodegenerative disease/cognitive impairment and foot/ankle pain and OA as well in the future. Second, the study is not free from selection bias, for example, footballers will not be selected randomly from all clubs but from our previous dataset. The same is true for the control population from the East Midlands because of the convenience of assessment in our local hospital. In addition, we are unable to randomly select participants who would like to have radiographs but need to take account of the distance to travel to the assigned NHS radiography departments. Third, individuals who have developed marked cognitive and motor impairment may be unable or decline to respond to the questionnaire, leading to selection bias towards milder conditions. Last but not least, the COVID-19 pandemic has affected the project and protocols. We were unable to *conduct* face-to-face assessments of cognitive and motor function, or to obtain radiographs (Nottingham, Southampton, Salford, Leeds and Imperial College London). The project was temporarily halted following the national lockdown due to the COVID-19 pandemic in March 2020. Following management committee meetings and advice from the University of Nottingham governance/Research Ethics Committee/Health Research Authority teams, it was decided to amend the original protocol from in-person assessments to telephone assessments in order to enable the study to begin. This approach was not possible for some of the neurocognitive assessments, and these were replaced with the TICS-m.[36 37] The TICS-m uses questions from the Montral Cognitive Assessment (MOCA) and MMSE, with which it is highly correlated; it is a much shorter assessment and can be administered by telephone so is safer for all involved during the pandemic and lockdown. The VFT, HVLT and IADL[41 43 45 46] were also retained as they can be administered by telephone;

however, the planned clinical neurological assessments (motor function) were removed.

Although the gold standard for testing cognitive function, and diagnosis of MCI and dementia is face-to-face assessment using a battery of standardised and validated cognitive tests,[55] these are not without their drawbacks.[56] TICS-m is the most frequently used telephone-based cognitive screening test in medium–large studies and epidemiological surveys.[55] Key advantages include little evidence of ceiling and/or practice effects, and its better acceptance by participants who find the telephone interview less threatening than a face-to-face battery of cognitive tests in clinic.[36] Although hearing and communication may pose an issue for some, this can be mitigated by a skilled interviewer.[36 57] In contrast, the foot/ankle radiographs require NHS clinical radiology services; these have to be fully suspended until research activities within Hospital trusts are resumed.

In conclusion, we expect that the results of our study will establish prevalence of foot and ankle OA symptoms; radiographic prevalence of foot and ankle OA; and prevalence of cognitive impairment and neurodegenerative diseases in ex-professional footballers compared with general population controls. Thus, providing the much-needed answers following on from recent evidence that professional footballers had a 3.5 times higher death rate from neurodegenerative diseases compared with population controls.[19] Given that football is played professionally and recreationally in over 200 countries by more than 250 million people,[58] robust evidence on mental and physical health and well-being in retired professional footballers are a vital first step in establishing risk profiles, which will assist in developing effective prevention strategies for mitigating against long-term health consequences.

## ETHICS AND DISSEMINATION

All study aspects were approved by the East Midlands-Leicester Central Research Ethics Committee and HRA (REC ref: 19/EM/0354) on 23 January 2020. Study results will be submitted to the FA, versus Arthritis, regulatory authorities, and peer reviewed journals for publication and presented at national and international conferences. Study participants will be provided with any resulting publications at their request.

**Author affiliations**
[1]Academic Rheumatology, School of Medicine, University of Nottingham, Nottingham, UK
[2]Centre for Sport, Exercise and Osteoarthritis Versus Arthritis, University of Nottingham, Nottingham, UK
[3]Population Health Sciences, University of Bristol, Bristol, UK
[4]NCSEM, School of Sport, Exercise and Health Sciences, Loughborough University, Loughborough, UK
[5]Sports Medicine, Nottingham University Hospitals NHS Trust, Nottingham, UK
[6]Colin Fuller Consultancy Ltd, Loughborough, UK
[7]Centre for Urgent and Emergency Research, The University of Sheffield, Sheffield, UK
[8]Psychology, University of Nottingham, Nottingham, UK
[9]Neuroscience@Nottingham, University of Nottingham, Nottingham, UK

**Acknowledgments** We are grateful to Mr Les Bradd and all other retired professional footballers who gave their time and expertise to contribute to this study, including those from Notts County Football Club, The PFA, especially Mr Richard Jobson, and all our PPI group. We also extend our gratitude to Dr Jerry M Brown for his helpful discussion and invaluable advice on the TYM test, and our collaborators Professors Cathy Bowen, and Terry O'Neill, and Drs Richard Wakefield, Mo Aslam, and Lucy Gates. We are indebted to Dr Charlotte Cowie for her guidance and support throughout the past few years. We wish to acknowledge the FA, the PFA, vs Arthritis, and the University of Nottingham for supporting this study, and King Abdulaziz University, Jeddah, Saudi Arabia for sponsoring AT's PhD programme. Finally, we are thankful to our participants both from the retired professional football players and those from the general population dwelling adults from the East-Midlands. Without their pertinent and persistent support over the past years, we would not be able to undertake this study especially during the COVID-19 pandemic/lockdown.

**Contributors** SE drafted, revised, finalised the manuscript, and implemented (conducted) the study and NHS Trust collaborating centres. SE developed and designed the FOCUS Footballer Questionnaire and data collection tools. GF, MD, WZ conceptualised the study. All authors were involved in the design and development of the study protocols. SE, AT, EH and WZ collected data. All authors reviewed the draft and approved the final manuscript. WZ is the guarantor for the project.

**Funding** This work is supported by grants from Versus Arthritis, grant number (21595) and The Football Association (FA), grant number (not applicable).

**Competing interests** CF provides consultancy services to the English Premier League.

**Patient and public involvement** Patients and/or the public were involved in the design, or conduct, or reporting, or dissemination plans of this research. Refer to the Methods section for further details.

**Patient consent for publication** Not applicable.

**Provenance and peer review** Not commissioned; externally peer reviewed.

**ORCID iDs**
Shima Espahbodi http://orcid.org/0000-0001-6293-1562
Gwen Fernandes http://orcid.org/0000-0003-0203-7053
Gordon Fuller http://orcid.org/0000-0001-8532-3500

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
