## [Reviewer comments · BMJ Open]

ARTICLE DETAILS

TITLE (PROVISIONAL)	Foot and ankle Osteoarthritis and Cognitive impairment in retired UK Soccer players (FOCUS): protocol for a cross-sectional comparative study with general population controls
AUTHORS	Espahbodi, Shima; Fernandes, Gwen; Hogervorst, Eef; Thanoon, Ahmed; Batt, Mark; Fuller, Colin; Fuller, Gordon; Ferguson, Eamonn; Bast, Tobias; Doherty, Michael; Zhang, Weiya

VERSION 1 – REVIEW

REVIEWER	Garrett Bullock University of Oxford, Nuffield Department of Orthopaedics, Rheumatology and Musculoskeletal Sciences
REVIEW RETURNED	21-Aug-2021

GENERAL COMMENTS	General Comments This study protocol aims to assess foot/ankle OA and neuro-cognitive impairment risk in former professional footballers compared to age and sex matched controls. Overall this is a very thorough and detailed protocol. It is well written and has quality statistical analyses. There are some clarifying points which are detailed below. Thank you for allowing me to review this article. Abstract Ideal power for a case control is 3:1 (maybe 4:1) ratio between controls and cases. Why a 1.2:1 ratio? This seems a bit arbitrary? Should note this may be better illuminated in the methods section. Line 21-22: This is extremely surprising this is not recognized as an occupation lifelong hazard in this work population. Introduction Line 4-5. Suggest adding the term prevalence to define percentages of thigh, foot/ankle, and knee injuries for better clarity. Line 39-46: I commend the authors for explaining the reasoning for including both foot/ankle and neuro cognitive impairments within the same study. I was wondering the reasoning behind this. Further, I commend the authors for creating a questionnaire where neuro cognitive questions are embedded in the questionnaire, for designed improved response rate. Methods
---

	I commend the authors for using a general population cohort from the East Midlands. From the title of the cohort, I was initially thinking there may be some berkson's bias with this cohort, especially since this is an OA study, however, I commend the authors for the reference, which gave much more detail on the generalizability of this cohort sample. Further, from your referencing to the BMC MSK Disorders protocol, it states that the survey invitation was sent to the 40,000 potential participants. While at least half of these would be female, and after non-response, and also marking of willingness to receive future research enquiries, is there more than 1100 available matching potential participants? I was wondering the reasons for an almost 1:1 matching ratio, instead of a 3:1 or 4:1, as this would be ideal power. Please provide clarity on this study design choice. Further, I think some of this can be clarified through a flow chart, specifically getting to 1100 matching participants (if this is all the cohort can supply for matching, instead of 3:1 etc.). Detailing initial recruitment cohort sample size, participated in the cohort, detailed for further research enquiry, and number available for age matching. Can you provide greater detail on randomization of calls? Overall: I commend the authors on the detail in reporting their questionnaire. RAND 36 is a long HRQoL questionnaire. Wondering the reason for RAND36 instead of VR-12, which is also free and has high validity? Reason for this question is if through piloting and through patient public involvement, time to completion and questionnaire drop out (at different questionnaire intervals) was assessed, as length of questionnaire, especially with longer validated PROMS can detract from participation. I commend the authors for the use of patient public involvement For social status, does this mean socioeconomic status (sorry this might be an American v UK terminology issue)? How will this be assessed (Household income?) This may have less response compared to other data points. How will this be accounted for. Extra space before linear. For risk factors, will this be univariable or will confounders be controlled for. If so what will they be, please clarify. I commend the authors for a very well done a priori sample size calculation. Discussion I commend the authors for giving a detailed account of the strengths and limitations of this project, along with the tweaking needed to be performed due to COVID-19.
--	--

REVIEWER	Masahiko Saito Seikeikai Chiba Medical Center, Orthopaedic Surgery
REVIEW RETURNED	09-Oct-2021

GENERAL COMMENTS	This paper shows the protocol of the study aiming to examine the
--

	prevalence and associated risk factors of foot/ankle OA and neurodegenerative disease in former professional footballers compared to matched general population controls. This is an interesting and essential research subject in term for clinician who treats professional footballers. While I consider the protocol of this study was well designed and easy to understand, I would suggest a minor revision considering the comments below before it can be accepted for publication. [abstract] Good overview of the paper. Clear and concise. [Background] Good overview although there seems to be a little lack of depth in this part. There is little lack of evidence and explanation for simultaneously examining two different pathological conditions, i.e. ankle/foot OA and cognitive impairment. It would be reader friendly if more details are introduced about investigating ankle OA and cognitive impairment in the same study. [Purpose] Clear. [Method] Good description of methods. It would be interesting if the authors can investigate the association between Ankle/Foot OA and cognitive impairment. [Discussion] Good discussion. I hope that my comments would be very helpful for the improvement of the manuscript.
--	--

VERSION 1 – AUTHOR RESPONSE

Reviewer: 1

Dr. Garrett Bullock, University of Oxford

Comments to the Author:

General Comments

This study protocol aims to assess foot/ankle OA and neuro-cognitive impairment risk in former professional footballers compared to age and sex matched controls. Overall this is a very thorough and detailed protocol. It is well written and has quality statistical analyses. There are some clarifying points which are detailed below.

Thank you for allowing me to review this article.

Abstract

Ideal power for a case control is 3:1 (maybe 4:1) ratio between controls and cases. Why a 1.2:1 ratio? This seems a bit arbitrary? Should note this may be better illuminated in the methods section.

Response: These participants (1100 controls and 900 cases) were recruited from a previous study (Fernandes et al Br J Sports Med 2018;52(10):678-83, Parekh et al Clin J Sports Med 2021;31:281-288) where individuals had consented to being contacted in the future about further studies related to general health and well-being. Therefore, based on written, informed consent, all eligible participants were invited to partake via this postal questionnaire.

Line 21-22: This is extremely surprising this is not recognized as an occupation lifelong hazard in this work population.

Response: We agree with the reviewer, hence the need to provide a robust evidence-base into this important topic in former professional footballers, given the recent findings of both increased risk of knee OA (Fernandes et al Br J Sports Med 2018;52(10):678-83) and neurodegenerative conditions related deaths (Mackay et al NEJM 2019;381:1801-8).

Introduction

Line 4-5. Suggest adding the term prevalence to define percentages of thigh, foot/ankle, and knee injuries for better clarity.

Response: Thank you. This has now been added (lines 4-5).

Line 39-46: I commend the authors for explaining the reasoning for including both foot/ankle and neuro cognitive impairments within the same study. I was wondering the reasoning behind this. Further, I commend the authors for creating a questionnaire where neuro cognitive questions are embedded in the questionnaire, for designed improved response rate.

Response: Thank you for the positive comments.

Methods

I commend the authors for using a general population cohort from the East Midlands. From the title of the cohort, I was initially thinking there may be some Berkson's bias with this cohort, especially since this is an OA study, however, I commend the authors for the reference, which gave much more detail on the generalizability of this cohort sample. Further, from your referencing to the BMC MSK Disorders protocol, it states that the survey invitation was sent to the 40,000 potential participants. While at least half of these would be female, and after non-response, and also marking of willingness to receive future research enquiries, is there more than 1100 available matching potential participants? I was wondering the reasons for an almost 1:1 matching ratio, instead of a 3:1 or 4:1, as this would be ideal power. Please provide clarity on this study design choice.

Response: To clarify, there are no more than 1100 control participants as mentioned above. We used the total number of participants available from the previous study who had indicated willingness to be contacted for future studies (900 footballers and 1100 controls). The maximum available pool of participants have been contacted and therefore the resulting ratio is 1.2:1.

Further, I think some of this can be clarified through a flow chart, specifically getting to 1100 matching participants (if this is all the cohort can supply for matching, instead of 3:1 etc.). Detailing initial recruitment cohort sample size, participated in the cohort, detailed for further research enquiry, and number available for age matching.

Response: We didn't match by age but contacted all available male participants who would like to take part in this further research study. As both cohorts were recruited at similar age bands, ie, 40 or more (Fernandes et al BMC Musculoskelet Disord 2017;18(1):404, Fernandes et al Br J Sports Med 2018;52(10):678-83) the mean age and range ended up being very similar.

Can you provide greater detail on randomization of calls? .

Response: (Line 28-29). It was not possible to randomize calls as originally intended due to the project launch in August 2020, in the midst of the COVID-19 pandemic. We did not know how the pandemic would affect response rates to a questionnaire study. As a result, it was decided that the most effective course of action would be to contact all interested and consenting participants, (in

chronological order) for a telephone assessment. This has been amended in the manuscript (lines 28-29).

Overall: I commend the authors on the detail in reporting their questionnaire.

RAND 36 is a long HRQoL questionnaire. Wondering the reason for RAND36 instead of VR-12, which is also free and has high validity? Reason for this question is if through piloting and through patient public involvement, time to completion and questionnaire drop out (at different questionnaire intervals) was assessed, as length of questionnaire, especially with longer validated PROMS can detract from participation.

Response: The RAND 36 was used in the previous football study and so kept here to be able to compare follow-ups at later time-points and for consistency of data collection. This was the general population health and well-being questionnaire that has been extensively used in the literature (Ware and Sherbourne Med Care 1992; 30(6):473-83) and in our own work (Fernandes et al Br J Sports Med 2018;52(10):678-83, Parekh et al Clin J Sports Med 2021;31:281-288). Hence, the study team chose to use RAND 36.

I commend the authors for the use of patient public involvement

Response: Thank you for the positive feedback.

For social status, does this mean socioeconomic status (sorry this might be an American v UK terminology issue)? How will this be assessed (Household income?) This may have less response compared to other data points. How will this be accounted for.

Response: Thank you for this. To clarify, we will use postcode as a proxy measure of socioeconomic status, which can tie in with English Indices of Multiple Deprivation (IMD) status in the UK and is widely used surrogate for socioeconomic status (<https://imd-by-postcode.opendatacommunities.org/imd/2019>). This has now been amended (lines 36-37).

Extra space before linear.

Response: Thanks, this has now been changed (lines 36-38).

For risk factors, will this be univariable or will confounders be controlled for. If so what will they be, please clarify.

Response: Yes, this is described under the Outcome measures and statistical analysis sub-section in our Methods and Analysis section. The confounders include age, Education, BMI, IMD, and these have been described in the methods (lines 35-38) and will be subsequently adjusted for in our analyses.

I commend the authors for a very well done a priori sample size calculation.

Response: Thanks for the positive comment.

Discussion

I commend the authors for giving a detailed account of the strengths and limitations of this project, along with the tweaking needed to be performed due to COVID-19.

Response: Thanks for the positive comment.

Reviewer: 2

Dr. Masahiko Saito, Seikeikai Chiba Medical Center

Comments to the Author:

This paper shows the protocol of the study aiming to examine the prevalence and associated risk factors of foot/ankle OA and neurodegenerative disease in former professional footballers compared to matched general population controls. This is an interesting and essential research subject in term for clinician who treats professional footballers. While I consider the protocol of this study was well designed and easy to understand, I would suggest a minor revision considering the comments below before it can be accepted for publication.

[abstract]

Good overview of the paper. Clear and concise.

[Background]

Good overview although there seems to be a little lack of depth in this part.

There is little lack of evidence and explanation for simultaneously examining two different pathological conditions, i.e. ankle/foot OA and cognitive impairment. It would be reader friendly if more details are introduced about investigating ankle OA and cognitive impairment in the same study.

Response: We thank the reviewer for this point. As explained in our manuscript (page 6, line 39-46), we chose this method of evaluating cognition alongside musculoskeletal pathology in part to increase the response rate in this specific group of participants. Our previous experience (PPI) found that often questions about mental health and well-being are interpreted with sensitivity and caution and may affect the responses themselves as well as the overall response rate to the study. As a result, we decided to evaluate general health and well-being, focusing on ankle and foot pain and osteoarthritis together with some questions on memory and cognition in order to provide a balanced approach to questionnaire design and simultaneously, to work towards a positive response rate (>50%). This was a follow up study to further characterise OA on other highly vulnerable joints – foot and ankle for footballers. It is also a new study to address the current public concern of the risk of dementia in professional football players because of head injury and concussion. Combining both in one study is more efficient, more representative (less influenced by personal interests in a specific condition) and possibly greater response rate (less sensitive and cautious for a specific condition).

[Purpose]

Clear.

[Method]

Good description of methods. It would be interesting if the authors can investigate the association between Ankle/Foot OA and cognitive impairment.

Response: Thank you

[Discussion]

Good discussion.

I hope that my comments would be very helpful for the improvement of the manuscript.

Response: very helpful and many thanks indeed!

VERSION 2 – REVIEW

REVIEWER	Masahiko Saito Seikeikai Chiba Medical Center, Orthopaedic Surgery
REVIEW RETURNED	16-Jan-2022
GENERAL COMMENTS	The authors have made comments and corrections to each of the Reviewer's comments, and I recommend that this paper is accepted.